# Dependance of Gauge Factor on Micro-Morphology of Sensitive Grids in Resistive Strain Gauges

**DOI:** 10.3390/mi13020280

**Published:** 2022-02-10

**Authors:** Yinming Zhao, Zhigang Wang, Siyang Tan, Yang Liu, Si Chen, Yongqian Li, Qun Hao

**Affiliations:** 1School of Optics and Photonics, Beijing Institute of Technology, No.5 South Zhongguancun Street, Haidian District, Beijing 100081, China; zhaoyinming@cimm.com.cn (Y.Z.); qhao@bit.edu.cn (Q.H.); 2Beijing Changcheng Institute of Metrology & Measurement, Beijing 100095, China; chensi@cimm.com.cn; 3Key Laboratory of Micro/Nano Systems for Aerospace of Ministry of Education, Northwestern Polytechnical University, Xi’an 710072, China; zgwang@mail.nwpu.edu.cn (Z.W.); sytan@mail.nwpu.edu.cn (S.T.); ly201169@mail.nwpu.edu.cn (Y.L.)

**Keywords:** resistive strain gauge, gauge factor, sensitive grids, micro-morphology, force sensor

## Abstract

The effect of micro-morphology of resistive strain gauges on gauge factor was investigated numerically and experimentally. Based on the observed dimensional parameters of various commercial resistive strain gauges, a modeling method had been proposed to reconstruct the rough sidewall on the sensitive grids. Both the amplitude and period of sidewall profiles are normalized by the sensitive grid width. The relative resistance change of the strain gauge model with varying sidewall profiles was calculated. The results indicate that the micro-morphology on the sidewall profile led to the deviation of the relative resistance change and the decrease in gauge factor. To verify these conclusions, two groups of the strain gauge samples with different qualities of sidewall profiles have been manufactured, and both their relative resistance changes and gauge factors were measured by a testing apparatus for strain gauge parameters. It turned out that the experimental results are also consistent with the simulations. Under the loading strain within 1000 μm/m, the average gauge factors of these two groups of samples are 2.126 and 2.106, respectively, the samples with rougher profiles have lower values in gauge factors. The reduction in the gauge factor decreases the sensitivity by 2.0%. Our work shows that the sidewall micro-morphology on sensitive grids plays a role in the change of the gauge factor. The observed phenomena help derive correction methods for strain gauge measurements and predict the measurement errors coming from the local and global reinforcement effects.

## 1. Introduction

Strain gauges are commonly used in aerospace, transportation, automotive industry, civil engineering, and even in the medical field. Based on a combination of elastic elements and strain gauges, the physical quantities acting on structures, such as strain, displacement, bending, torque, and acceleration, can be measured directly or indirectly [1,2,3].

The strain gauge incorporates the resistance change linearly with deformation in its sensitive grids [1]. During the measurement process, strain gauges sense the elastomeric strain and translate it into a change of resistance value. The term to describe the correlation between these two variables is called gauge factor (GF), which is defined as the ratio of the relative resistance change (RRC) to the strain [4]:
(1)$$GF=\frac{\Delta R/R}{\varepsilon }$$
where the ΔR/R is RRC in strain gauges, and ε is the corresponding strain.

When strain is transmitted from the elastic element to the strain gauge, the strain reduction is caused by a mismatch in stiffness between the sensitive grid and the tested sample [3], and so do strain distortions on the edges [5,6]. Compared with strains experimented by elastic elements in the absence of strain gauges, those measured with strain gauges have lower values. This strain reduction and redistribution in resistive strain gauges contribute to the known reinforcement effect [7,8] which can lead to significant errors in determining the GF.

The structure configuration of the resistive strain gauge has a profound effect on electrical conductivity, which results in a varying GF of the strain gauge. The surface topographies from three typical metallic materials have been quantitatively characterized and analyzed to investigate the effect of micro-stress concentrations within the sensitive grids [7]. The integrated influences of characteristic surface parameters on micro-stress behavior have been considered to estimate the impact of surface conditions on the performance of GFs. The geometrical dimensional sizes of the strain gauge determine the GF and the measurement uncertainties due to strain disorder distributions [9,10]. It has been confirmed that the thinner film has a higher value of GF [11], and that the increased island sizes of sensitive grids decrease the GF [12,13]. The optimized end structures and corner arcs of susceptible grids reduce the transverse effect and improve their GFs [14].

In fact, strain gauges deflected from the perfect cross-section and the ideal sidewall profile of sensitive grids have defects in creep-lag effect, long-term stability and temperature self-compensation [15]. However, the sensitive grid patterns are usually fabricated by the processes of lithography and wet etching, whose poor selectivity of the damp etching results in irregular sidewall micro-morphology on sensitive grids [6]. These defects induce the electrical conductivity deviation of resistive strain gauges. By now, few works of literature focus on correlations between the property of the GF and the micro-morphology on the sidewall profiles. In order to calibrate the GF obtained by strain gauges during measurements and improve the long-term stability of resistive strain sensors, we observed the micro-morphology on various commercial strain gauges and calculated the geometrical parameters of the sidewall profiles. The finite element and experiment method had been used to investigate the GF determined by strain transfer characteristics of the strain gauges. Thereby, the focus has been laid on the sidewall micro-morphology of the sensitive grids, which would change the stress–strain characteristics of resistive strain gauges.

## 2. Geometrical Model and Simulations

For specific manufacturing equipment and processes, the micro-morphology on sensitive grids of strain gauges obtained by wet etching is controllable within a certain tolerance. However, it is this varying degree of tolerance that can result in strain gauges of very different qualities from one manufacturer to another. In detail, this difference mainly lies in the roughness of the sidewall profile on sensitive grids. To have a visual and analytical understanding, the sidewall micro-morphology of a typical strain gauge have been observed by confocal microscopy (KH-7700 by HIROX company in Shanghai, China). Additionally, the geometric characteristics of the sensitive grids were obtained through a scanning electron microscopy (SEM). As shown in Figure 1a, the shallow gray area in the SEM image refers to sensitive grids, while the dark gray is the blank gap between them. With image processing methods, Figure 1b,c show the sidewall profile of a sensitive grid from the viewpoint of a binary image.

In this study, geometrical parameters of the sidewall profile on sensitive grids include average profile unit width $R_{a}$, average profile unit length $R_{sm}$, the maximum width $R_{ai}$ and maximum length $R_{smi}$ of each unit. For a certain sidewall profile line, the $R_{a}$ is defined by: (2)$$R_{a}=\frac{1}{m}\sum_{i=1}^{m}\left|R_{ai}\right|,$$
where *m* is the total number of discrete points. Similarly, the $R_{sm}$ is written as:(3)$$R_{sm}=\frac{1}{n}\sum_{i=1}^{n}R_{smi},$$
where n is the total number of the profile units. These definitions describe the amplitude and frequency of rough sidewall profiles. 

By random observation and measurement, we obtained the above geometrical parameters of the sidewall profiles from four kinds of commercial resistive strain gauges, and the number of which were dozens of. Here we need to add that, after further research [16], we have demonstrated that textures left on the upper surfaces of sensitive grids due to resistance trimming also affect the performance of strain gauges, and in this study, in order to control the study variables, we chose commercial strain gauges without significant upper surface textures. Based on the data obtained, we classified these strain gauges according to their sidewall qualities and grouped them into a total of four categories. Additionally, we selected the most representative binarized image from each of these four categories and obtained the geometrical parameters for each group by averaging, as shown in Table 1, their geometrical parameters vary. From the horizontal view of Table 1, $R_{a}$, $R_{ai}$ and $R_{sm}$ increase sequentially. Moreover, the $R_{a}$, as the arithmetic mean of $R_{ai}$, is the smallest among the three. From the longitudinal view, the first image shows a smooth sidewall and smaller geometrical parameters, whereas Sample 4 takes on a rough and worse condition of sensitive grids. All three sets of values gradually increase as the sidewall profile becomes progressively rougher from the first to the fourth image. When the sidewalls are rough to a certain degree, $R_{ai}$ and $R_{sm}$ can even reach the order of tens of microns. 

Among the above four kinds of resistive strain gauges, the basic dimensional sizes corresponding to the type HBM1-DY13-3/350 were selected to build the simulation model, in which the grid width is 50 μm and the thickness is 5 μm [17]. Following the parameters in Table 1, a geometrical model of the sidewall profile was constructed to study the GF. As shown in Figure 2, the complex waveforms or zigzag sidewalls demonstrated in Table 1 are simplified to be a standard cosine curve, where the amplitude (A) and period (T) describe the cosine type profile. In real work situations, the strain in elastic elements produced by an applied tension force is passively transferred to the sensitive grids through the substrate and adhesive layer [14], to apply a uniform shear force into the sensitive grid and reduce the reinforcement effect, the sizes of both substrate and adhesive layer are set to be larger than that of the sensitive grid.

In addition, the Poisson’s ratio of each elastic modulus and material’s layer plays a role in GF; however, since we chose fixed materials, only the sidewall defects were mainly investigated here. 

The piezo-resistance property of metal materials is defined as the bulk resistivity change due to applied stress or strain. The resistivity change induced by stress in cubic symmetry materials is expressed as below general form [18]:(4)$$\left[\begin{array}{c} \Delta \rho_{1}/\rho_{0}\\ \Delta \rho_{2}/\rho_{0}\\ \Delta \rho_{3}/\rho_{0}\\ \Delta \rho_{4}/\rho_{0}\\ \Delta \rho_{5}/\rho_{0}\\ \Delta \rho_{6}/\rho_{0} \end{array}\right]=\left[\begin{array}{cccccc} \pi_{11} & \pi_{12} & \pi_{12} & 0 & 0 & 0\\ \pi_{12} & \pi_{11} & \pi_{12} & 0 & 0 & 0\\ \pi_{12} & \pi_{12} & \pi_{11} & 0 & 0 & 0\\ 0 & 0 & 0 & \pi_{44} & 0 & 0\\ 0 & 0 & 0 & 0 & \pi_{44} & 0\\ 0 & 0 & 0 & 0 & 0 & \pi_{44} \end{array}\right]\left[\begin{array}{c} \sigma_{1}\\ \sigma_{2}\\ \sigma_{3}\\ \sigma_{4}\\ \sigma_{5}\\ \sigma_{6} \end{array}\right]$$
where $\rho_{0}$ is the initial resistivity, $\Delta \rho_{i}/\rho_{0}$ is the resistivity change, $\sigma_{i}$ is the stress, and $\pi_{ij}$ is the piezo-resistance coefficient. The properties of the constantan material used for the strain gauge in our calculations are summarized in Table 2, which are consistent with the data in [18,19].

The parameters of both amplitude (A) and period (T) in the cosine profile model (Figure 2) are normalized by the widths of the sensitive grids (W). Numerically, they are expressed as A/W (relative amplitude, RA) and T/W (relative period, RP), respectively. On this basis, RRC and GF were calculated in varying sidewall profiles and external stress conditions. In Figure 3a, RRC related with RA and RP were presented. In the interested strain range of 0 to 2000 μm/m, RRCs are almost linearly proportional to the applied strain. For a certain strain (1000 μm/m), the growing tendency of RRC resulting from the increase of RA and RP can be easily told in the partial enlarged view (Figure 3a). This phenomenon is well confirmed by the calculation results in Figure 3b. Particularly, it is worth noting that the scale of the abscissa interval in Figure 3b is much larger than the $0.06\times 10^{-4}$ variation range of the ordinate; therefore, the actual curves are very close to horizontal straight lines, which means GFs approximately maintain constant. As for the small-scale increase in GFs caused by RA and RP, we can understand it as: the stress redistribution affected by the sidewall profile will aggravate the creep effect of the material [16], which eventually affect GF. From above, the varying sidewall micro-morphology of sensitive grids leads to changes in both RRC and GF. Consequently, the deviations between the rough sidewall profile and perfectly smooth one should be taken into account so as to improve the measurement accuracy of the sensitive grids [7,20]. 

As shown in Figure 4, increasing RA (A/W) or RP (T/W) results in growing RRC. In the case of a constant RP, Figure 4a shows a sharp rising of RRC over the increasing RA, and this variation in RRC is nonlinear over a tenth of the sensitive gate width, meaning that the amplitudes of sidewall profiles gradually amplify the deviation of relative resistance change in the strain gauge under a constant load. However, as shown in Figure 4b, the results are much different. For a given RA, the RRCs fluctuate in a small range around a certain value. Take A/W = 4% for example, the waveform length would almost not lead to relative resistance change, and this is in line with the theory [21]. Still, the volatility of the RRC at the beginning step of the curve is more pronounced, while it flattens out more in the rest section. It is not difficult to understand, with a fixed RA, an increasing RP represents a smoother sidewall profile of the sensitive grid that possess a steady RRC.

These phenomena seem reasonable when considering the correlation between sidewall roughness and stress deformations. As stated above, when RP becomes long enough, the sidewall profile equally becomes smooth. The smooth profile would reduce the local strain concentration until a homogeneous strain distribution forms. Under this condition, the deformation in resistive strain gauges would represent the strain in elastic elements to the greatest extent, and the measured GF is a standard one. Yet, there is no way to produce strain gauges with absolutely smooth sidewalls on sensitive grids. As the non-ideal sensitive grid is subjected to external stress, the deformation of each aera is uneven and the concave parts produce local stress concentrations. As shown in Figure 5, when RAs are the same in Figure 5a,b, the longer RP (Figure 5b) reduces the maximum local strain concentration within the sharp dimensional area, which results in a relatively even strain distribution within the sensitive body. Further, the even distribution induces a much larger change of relative resistance, or a more accurate sensitivity. In Figure 5a, a sharper shape on the sidewall profile produces a worse local strain concentration than that in Figure 5b, and the lower strain distribution over the grid body decreases the resistance change and leads to measurement deviation. For the same reason, for an RA that is less than 4% of the grid width (Figure 4b), the RRC becomes a basically unchanged number even the RP range from 40% to 80% in our simulations, meaning that as long as the sidewall is relatively smooth, the RRC basically maintains at the standard constant no matter how high or low RA/RP is.

## 3. Experimental Results and Discussion

The micro-morphology of sensitive grids depends on the fabrication processes. Especially, during lithography, both mask and photoresist determine the quantity of fabricated pattern and the sidewall profile of sensitive grids. Figure 6 shows two process flows fabricating strain gauges, in which the discrepant sidewall profiles can be obtained. A Cr-mask and positive photoresist are used in the craft I, and a film mask and negative photoresist are adopted in the craft II. The former produces a smoother profile with finer resolution while the later brings out a rough one with micro-bulges on [22,23].

To have clear demonstration and contrast in the following experiment, two groups of strain gauges we used were manufactured by the above two processes, respectively, and there were dozens of samples per group. The structures of all these samples are the same (Figure 6) [6], and the materials of the main layers of the structure are shown in Table 3. Based on the principle of random measurement, it turned out that the quality of the sidewalls of the samples within the group were all about the same, while there were large differences between the two groups. The sidewall of the most representative sample in each group is shown in Figure 7b,c. Estimated with Equations (2) and (3), their geometrical parameters are listed in Table 4. In addition, we selected two more strain gauges with almost the same qualities of sidewall profiles from each group, that is, based on the three strain gauges in each group, we obtained their stress response and GF by the experiment.

All the six samples were measured in a testing apparatus for strain gauge parameters [17,24]. As shown in Figure 8c,d, in order to be identical to actual use, the strain gauges were glued to the surface of the rigid standard elastic beam and then solidified by thermal curing process. A standard applied force produces a uniform strain in the middle of the standard flexible beam. As shown in Figure 8a,b, the pressure on the power arms bends the elastic beam delivering both negative and positive strain to the standard elastic beam. There are equal strain zones in the center area of the standard elastic beam. The strain values are calculated by the deflection theory, in which the deflection displacement of f+ and f− are measured by a standard flexometer. In addition, a Wheatstone bridge for converting the strain into an electrical signal was connected to a data acquisition amplifier (HBM QuantumX MX1615B) [17].

The linear loading force Q provides an elastic beam strain ranging from 0 to 1000 μm/m. Two groups of strain gauge samples were measured and the strain responses are presented in Figure 9. In the initial region of small strain values, the measured values from two groups are almost indistinguishable. However, with the strain increasing up to 500 μm/m~1000 μm/m, the strain deviation in Group II increases in a small range compared to the Group I. Considering that commercial strain gauges are inherently high-precision components, this small difference in the measurement due to microscopic sidewall profiles still makes sense. Numerically speaking, using the data in Figure 9, the linearities of the two groups can be obtained using the least squares method and they are 1.005 (Group I) and 1.023 (Group II), respectively. The deviations of the linearities from standard value 1 are 0.5% and 2.3%, which results from the sidewall profile difference between the two groups of fabricated samples.

Moreover, within 1000 μm/m of the loading strain, we measured the GFs from the two groups of samples. In Figure 10, The results indicate that the strain gauges in Group I, which have smaller roughness of sidewall profiles, have larger GFs than those in Group II. Although the measurements were approximately linearly correlated between the measured strain gauges and the standard test apparatus, the data in Figure 10a,b show that the GF values for Group I are definitely greater than those for Group II, in other words, the strain gauge with a smooth sidewall profile have a better sensitivity. Among the measured data in Figure 10, the average GFs in two groups are 2.126 and 2.106, respectively, and the difference between these two values is 0.02. The reduction of the GF decreases the sensitivity by 2.0%. Furthermore, we have shown that the roughness of the textured surface on sensitive grids also shortens the fatigue lifetime and enhances the creep effect of strain gauges [23]. The deterioration of the GF leads to a great amount of measurement uncertainty against the long-term reproducibility in metrology force sensors. The experiment’s conclusion agrees well with the obtained simulation results shown in Figure 3 and Figure 4.

## 4. Conclusions

The dependence of the micro-morphology on resistance changes and gauge factors in resistive strain gauges were investigated numerically and experimentally. The dimensional sizes and their geometrical parameters of the sidewall profiles on sensitive grids are obtained utilizing scanning electron microscopic and the image processing techniques. Based on these data, a model of the sensitive grid with variable cosine profiles had been built and used in simulations, where cosine curves are approximately equivalent to the actual complex waveforms or zigzag sidewall profiles. Subsequently, the relative resistance changes and gauge factors under different amplitudes and periods of the cosine model have been analyzed by simulations. Additionally, using testing apparatus for strain gauge parameters, we tested two groups of strain gauges which were manufactured by two fabrication processes and have sensitive grids with different sidewall qualities. The results in both the simulation and experiments show that the periodic depression on the sidewall enhances the local strain concentration and weakens the strain distribution over the grid body, indicating that the rough micro-morphology leads to the decrease in gauge factors and results in the relative reduction of accuracy and sensitivity of strain gauges. These results make it clear that the rough sidewall profiles exacerbate to the measurement uncertainty. For ultra-high-precision sensors, such deviations are not allowed; therefore, the disadvantages caused by the micro-morphology of strain gauges must be taken into consideration, and the measurement must be calibrated in the best possible way.

## Figures and Tables

**Figure 1 micromachines-13-00280-f001:**
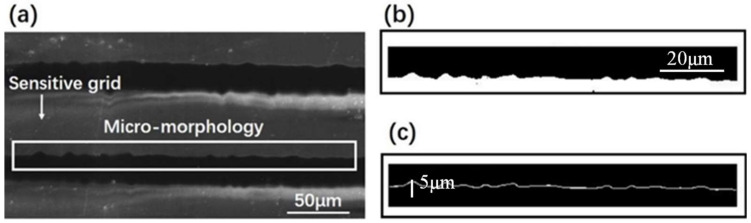
The microscopy image and the obtained sidewall micro-morphology of a typical strain gauge. (**a**) Scanning electron microscope image, (**b**) Binary image, and (**c**) Sidewall profile of a single sensitive grid.

**Figure 2 micromachines-13-00280-f002:**
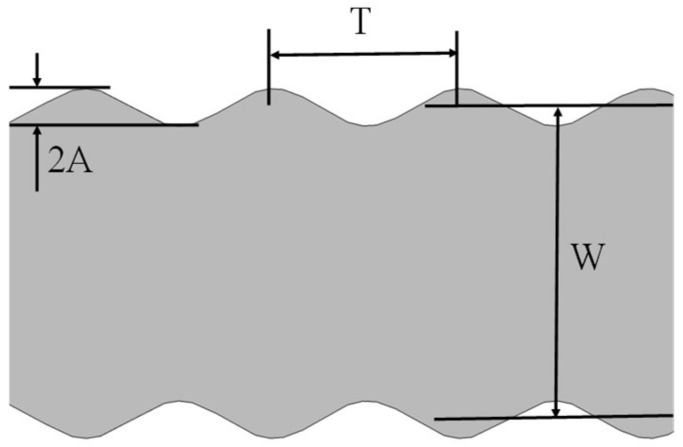
The geometry model for a segment of a sensitive grid in a strain gauge. The parameters of amplitude (A) and period (T) describe the cosine-curve profile. The width (W) of the sensitive grid is located between two dashed lines estimated by the least square methods.

**Figure 3 micromachines-13-00280-f003:**
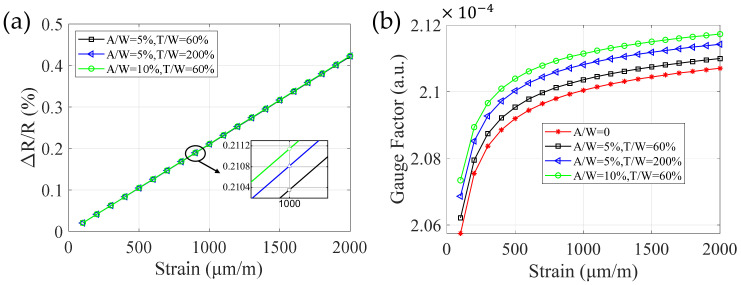
The dependence of RRC and GF on RA and RP in sensitive grids. (**a**) Three susceptible grids with different RA and RP are calculated and increasing RRCs were led by both. (**b**)The calculated GFs increase with the strain load in varying RA and RP conditions. Here, an ideal profile with A/W = 0 is used to be the reference for a clear demonstration in (**b**).

**Figure 4 micromachines-13-00280-f004:**
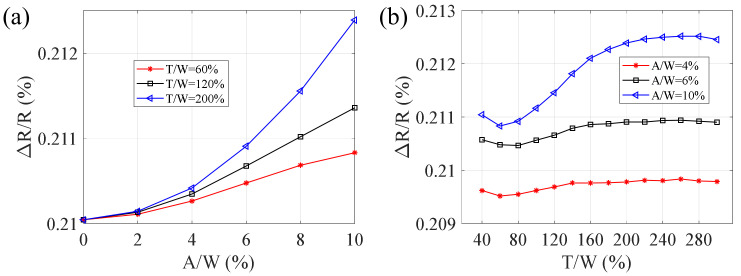
RRC and GF are affected by the geometrical parameters of (**a**) RP and (**b**) RA in the cosine profile. The loaded strain is 500 μm/m.

**Figure 5 micromachines-13-00280-f005:**
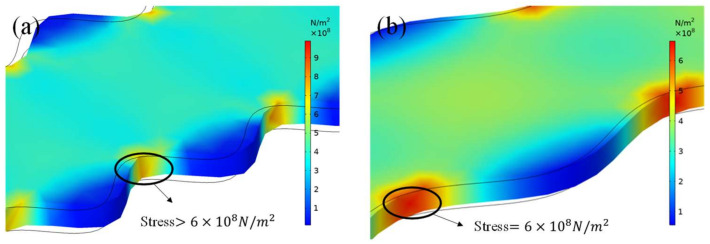
The strain distributions due to the sidewall profile roughness of the sensitive grids under the same load. The sidewall profile is assumed to be (**a**) T/W = 60% and (**b**) T/W = 120%.

**Figure 6 micromachines-13-00280-f006:**
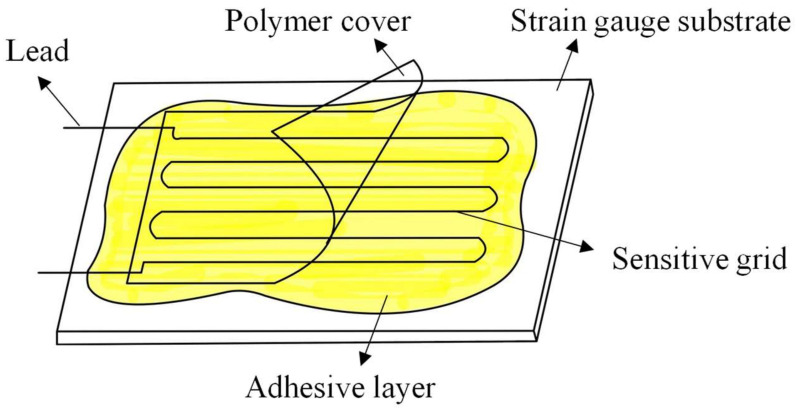
Structural diagram of the strain gauges used in the experimental test [6].

**Figure 7 micromachines-13-00280-f007:**
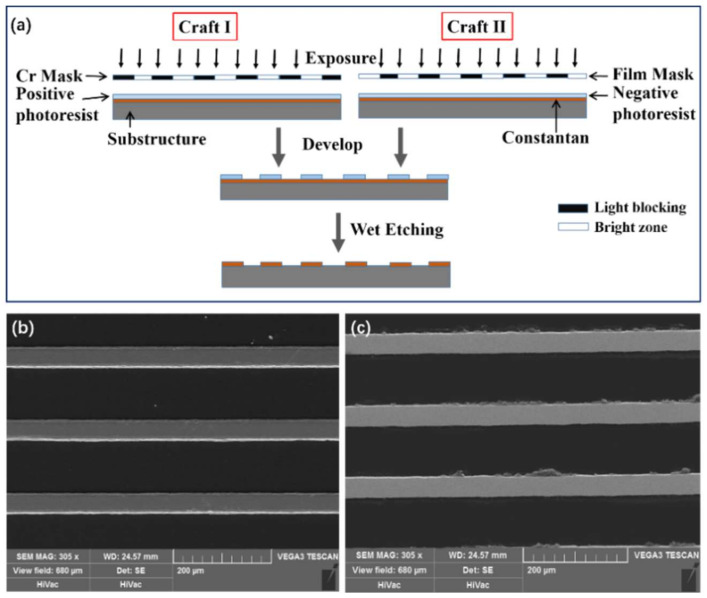
Fabrication processes for obtaining strain gauges and the resulting sidewall quantity. (**a**) Two fabrication processes. The SEM images of micro-morphology produced by (**b**) the Cr-mask and positive photoresist and (**c**) film mask and negative photoresist.

**Figure 8 micromachines-13-00280-f008:**
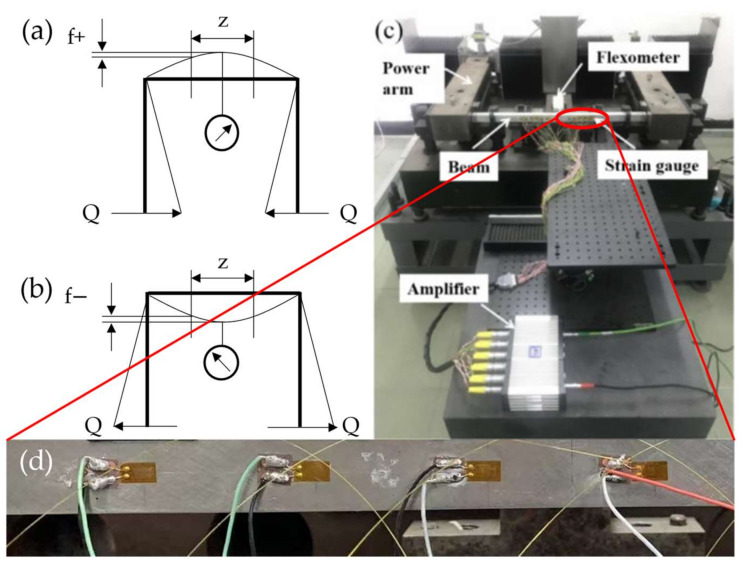
Testing apparatus for strain gauge parameters. A schematic diagram of the negative (**a**) and positive (**b**) strain produced in a standard elastic beam by a linear loading Q applied on the power arms. (**c**) The picture of the testing apparatus for strain gauge parameters. (**d**) Photo of a standard elastic beam with strain gauges glued.

**Figure 9 micromachines-13-00280-f009:**
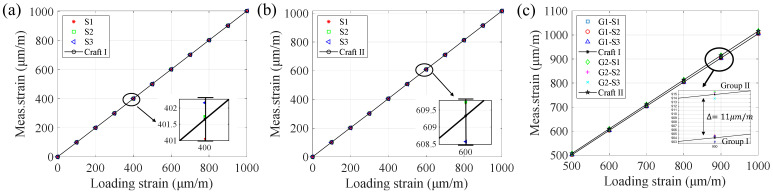
The strain response from two groups of samples to the standard strains. The geometrical parameters of (**a**) Group I and (**b**) Group II are listed in Table 3. (**c**) The parts of (**a**,**b**) in the range 500–1000 μm/m are plotted onto the same graph. The strain in the ordinate was measured by the samples, while the strain in the abscissa was calculated by the loading force and the displacements coming from the flexometer.

**Figure 10 micromachines-13-00280-f010:**
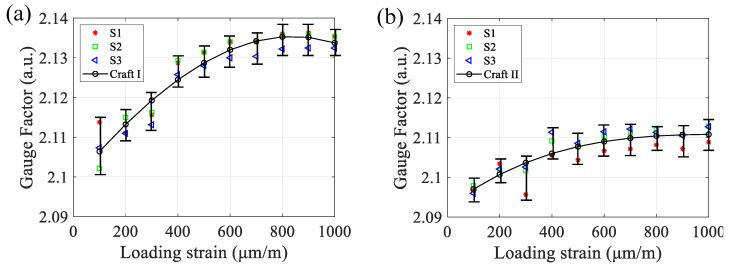
The GF was measured from two groups of fabricated strain gauges. Three samples with smoother sidewall profiles manufactured by Craft I (**a**), while the other three fabricated by Craft II have rougher sidewall profiles (**b**). The loading strain increases from 100 to 1000 μm/m.

**Table 1 micromachines-13-00280-t001:** Typical sidewall profile parameters of sensitive grids (Units: μm).

Serial Number	Binarized Image	*R_a_*	*R_ai_*	*R_sm_*
1	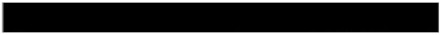	0.4	2.2	2.7
2	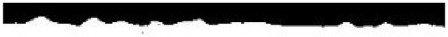	1.2	6.7	45.4
3	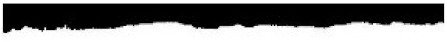	2.3	14.1	38.1
4	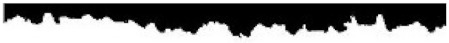	6.0	26.0	36.5

**Table 2 micromachines-13-00280-t002:** Material properties of constantan.

Density (kg/cm^3^)	Young’s Modulus (GPa)	Poisson’s Ratio	Resistivity (Ω·μm)	$\pi_{ij}\;(\times 10^{-12}\;{\mathrm{Pa}}^{-1})$
8.88	160	0.329	0.48	$\pi_{11}$$=2.3,\;\pi_{12}$$=2.3,\;\pi_{44}$ = 0

**Table 3 micromachines-13-00280-t003:** Materials of the main layers of strain gauges used in the experimental test.

Strain Gauge Substrate	Adhesive Layer	Sensitive Grids
Polyimide	Epoxy resin	Constantan alloy

**Table 4 micromachines-13-00280-t004:** The sidewall surface in two groups of fabricated strain gauge samples.

Sample	Craft	SEM Images	*R_a_*	*R_ai_*	*R_sm_*
Group I	Craft I	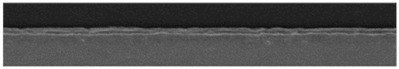	0.48	2.79	4.00
Group II	Craft II	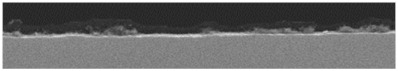	1.37	9.90	29.40

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
