# Peer review of "Dependance of Gauge Factor on Micro-Morphology of Sensitive Grids in Resistive Strain Gauges"

_micromachines, 2022, doi:10.3390/mi13020280_

Round 1

Reviewer 1 Report

The paper studies the effects of straining rate and micro-morphology to resistive metal strain gauges. In this work, the authors use strain gauges with two types of sidewall micro-morphologies differed in fabrication processes, and the relative resistance change and gauge factors were measured using specific calibration system. The author concluded that the relative resistance change tends to increase with rugged sidewall profiles. Even though the content is meaningful to some extent, there are required major revisions as below.

  1. It is not clear the specifications of the tested samples? What are the material constructing the strain gauges?
  2. What causes the differences in micro-morphology in the sidewall of those samples? Are they just randomly measured or are they controllable?
  3. Line 156: “…However, Figure 3 (b) indicates that the gauge factor increases exponentially to the strain for varying relative amplitude and sensitive grids…” is INCORRECT. The GFs does not increase exponentially with the strain! It actually saturates at higher strain.
  4. It is fundamentally incorrect : Data in Fig. 3a) and 3b) and contradicts each other: It is no doubt that ??=ΔR/?⁄?, according to Fig 3a) ΔR/? vs ? are in a linear relationship, therefore GF should be constant not varying as Fig. 3b). If this is the case, a comprehensive explanation needed here.
  5. Even though the experimental apparatus was shown in Fig. 7. It is still unclear how the sample are mounted on the strain induce structure (no indication of the sample on the photos)
  6. Considering the anisotropy of piezoresistive effect (Eq. 4), have any test with a different crystallographic orientation has been conducted (apart from the main orientation). What cause the strain anisotropy in metallic based materials which typically isotropic? Good references to for strain isotropy/anisotropy and testing method can be referred in: 10.1063/1.5037545, 10.1109/LED.2017.2726016
  7. There are many grammatical errors: “resistance strain gauges” -> “resistive strain gauges”, “sensitive coefficient” -> “sensitivity” etc. The English needed to be thoroughly revised.

Reviewer 2 Report

An interesting piece of work.  Some comments to improve the manuscript before publication.

  1. Can he authors provide information of how many samples were used?  As these work is statistically related, the number of samples is important to provide this information for Figure 3, 4 and 9.  Figures 4 and 5, should have the statistical information such as mean and the standard deviation similar to that of Fig 9.
  2. Figure 5, attempts to provide some scientific explanation of the observations.  However, it needs to be extended to comprehensively provide a theoretical analysis where it can be used to compare the experimental results.
  3. Is side-wall micromorphology the only factor that affect the gauge factor? Is there any other parameters and how do the authors assumed that all the commercially purchased sensors do have the other parameters to be controlled or similar?
  4. Figure 8a and b should be plotted on the same graph so that Craft I and II can be compared graphically.  Further with 3 samples in each Craft, it is premature to use the deviation as conclusion.  Similar to Figure 9, with only 3 samples, it is statistically weak.

Round 2

Reviewer 1 Report

The current revised version can be accepted.